# Study of Cutting Power and Power Efficiency during Straight-Tooth Cylindrical Milling Process of Particle Boards

**DOI:** 10.3390/ma15030879

**Published:** 2022-01-24

**Authors:** Rongrong Li, Qian Yao, Wei Xu, Jingya Li, Xiaodong (Alice) Wang

**Affiliations:** 1Co-Innovation Center of Efficient Processing and Utilization of Forest Resources, Nanjing Forestry University, Nanjing 210037, China; rongrong.li@njfu.edu.cn (R.L.); yq15905157072@163.com (Q.Y.); 2Anhui Product Quality Supervision & Inspection Research Institute, Hefei 230051, China; lijingya0202@163.com; 3Department of Wood and Forest Sciences, Laval University, Quebec City, QC G1V 0A6, Canada; xiaodong.wang@sbf.ulaval.ca

**Keywords:** cutting power, power efficiency, particle board, milling, response surface methodology

## Abstract

The cutting power consumption of milling has direct influence on the economic benefits of manufacturing particle boards. The influence of the milling parameters on the cutting power were investigated in this study. Experiments and data analyses were conducted based on the response surface methodology. The results show that the input parameters had significant effects on the cutting power. The high rake angle reduced the cutting force. Thus, the cutting power decreased with the increase in the rake angle and the cutting energy consumption was also reduced. The cutting power increased with the rotation speed of the main shaft and the depth of milling induced the impact resistance between the milling tool and particle board and the material removal rate. The *p*-values of the created models and input parameters were less than 0.05, which meant they were significant for cutting power and power efficiency. The depth of milling was the most important factor, followed by the rotation speed of the main shaft and then the rake angle. Due to the high values of R^2^ of 0.9926 and 0.9946, the quadratic models were chosen for creating the relationship between the input parameters and response parameters. The predicted values of cutting power and power efficiency were close to the actual values, which meant the models could perform good predictions. To minimize the cutting power and maximize the power efficiency for the particle board, the optimized parameters obtained via the response surface methodology were 2°, 6991.7 rpm, 1.36 mm for rake angle, rotation speed of the main shaft and depth of milling, respectively. The model further predicted that the optimized parameters combination would achieve cutting power and power efficiency values of 52.4 W and 11.9%, respectively, with the desirability of 0.732. In this study, the influence of the input parameters on the cutting power and power efficiency are revealed and the created models were useful for selecting the milling parameters for particle boards, to reduce the cutting power.

## 1. Introduction

Under the environmental protection goal of carbon neutrality and carbon peak, sustainable manufacturing has become the new development direction of the manufacturing industry. This has led to a focus on recycling and the conservation of natural resources and energy in furniture manufacturing processes [1,2,3]. Particle boards (PBs) as a kind of wood-based panel, are widely applied for the manufacturing of custom furniture in China [4,5,6] and milling occupies a large proportion of that manufacturing process [7,8]. Among the previous studies of material processing, most of them focused on revealing the influences of cutting parameters, cutting tools geometry and materials properties on the cutting tools’ conditions, cutting force, cutting efficiency and cutting qualities [9,10,11]. For PB processing, Boucher et al. studied the influences of the helix angle and density variation on the cutting force of PBs. The helix angle had positive influence on the decrease in cutting force and improvement of tool life [12]. In the peripheral up-milling of PBs, the higher the cutting speed used, the higher the principal cutting forces [13]. In addition, the properties of PBs also have influence on the cutting force. Bouzakis and Koutoupas found that the specific cutting force increased with the increase in the particle board’s mechanical strength critical stresses and the linear correlation between them was obvious [14].

The optimization of process parameters for reducing the final production costs is also one of the biggest challenges in the machining process. Hence, revealing the effects of processing parameters on the cutting power and power efficiency and achieving the optimized parameter combination would be meaningful for the furniture manufacturing industry [15,16]. At present, many scholars have focused on the influence of technical parameters on power consumption, modelling techniques for predicting and optimizing energy consumption, etc. [17,18,19]. From the previous research studies, the milling energy consumption and efficiency of machining processes for heat-treated wood has been found to be less than that for un-treated wood due to chemical degradation and reduced wood density. Additionally, the power consumption decreased as the temperature of heat treatment increased [20,21]. Normally, the specific cutting energy increases with the wood density, but decreases with the increase in moisture content [22]. In addition to the milling process, wood sawing power consumption has also been of interest. The influence of cutting motion parameters and saw blade parameters on power consumption were investigated in the past. Power consumption has been found to vary with different saw blades and increase under higher feed rates. With the increase in the feed speed, not only the sawing power increased, but also the waviness increased [23,24,25].

In addition to revealing the influencing rule of processing parameters on energy consumption, it is also meaningful to find the trade-offs between productivity, cut quality and power consumption. Different modelling techniques (such as artificial neural networks (ANNs), deep learning (DL), machine learning (ML), etc.) have been studied to monitor tool conditions, reduce cutting power consumption and increase energetic efficiency [26,27,28,29,30,31]. Tiryaki et al. employed artificial neural networks to minimize the surface roughness and power consumption of wood abrasive machining processes. Their results showed that the created models could give accurate prediction of power consumption and surface roughness as a result of low values of mean absolute percentage error [32]. De Melo et al. used the robust optimization technique to optimize the energy efficiency of the *Pinus taeda* wood planning process. Their results showed that the cutting motor rotation, feed motor rotation and cutting depth had significant influence on the specific cutting energy. They also found that the optimal parameter combination for lower power consumption was determined by the mean square error (MSE) [33]. The ANN modeling method is also a common modeling technique to predict the cutting power. During the wood planning process, the mean absolute percentage error values between the actual and predicted values for the training data set and testing data set were very small. The R^2^ values were close to the value of 1, which also meant that the ANN models could effectively be used in wood machining optimization to reduce the cutting power [34].

However, there is no published work focusing on revealing the effects of the milling parameters on the cutting power and power efficiency during the PB milling process. Moreover, the explicitly quantifying relationships between input parameters and response parameters have not been revealed either. In this paper, the cutting power and power efficiency during a straight-tooth cylindrical milling process of PBs were studied. The rake angle of the milling cutter, the rotation speed of the main shaft and the depth of milling were selected as input parameters. The response surface methodology (RSM) was applied to investigate the effects of the input parameters on the cutting power and power efficiency and establish the relationship between the input parameters and response parameters. These results will be beneficial in revealing the change tendency of the response parameters and guiding the selection of the input parameters to reduce the spindle power during the PB milling process.

## 2. Materials and Methods

### 2.1. Materials

Samples with the size of 150 mm × 80 mm × 18 mm (length × width × thickness) were cut randomly from commercial PB products. According to the Chinese standard GB/T 17657-2013 [35], the tests for modulus of rupture and modulus of elasticity of the supplied PBs were repeated five times and their mean values were 17.3 MPa and 1622 MPa, respectively. A density of 695.3 kg/m^3^ and a moisture content of 10.2 ± 2% were also documented.

### 2.2. Experimental Setup and Cutting Tools

A computerized numerical control (CNC) machining center was applied to carry out the milling experiments. The spindle power of the CNC machining center was measured by a three-phase power analyzer (AN87300, Ainuo Co., Ltd., Shandong, China). In this study, all the samples were cut by peripheral milling with a straight-tooth cylindrical milling tool. The milling tool had three inserted knives, which were made of cemented carbide. The diameter and wedge angle of the selected milling tool were held constant at 180 mm and 45°, respectively. The schematic diagram of the experimental setup is shown in Figure 1.

### 2.3. Methods

The RSM, with its preponderance of convenient modeling and strong ability to deal with multi-objective optimization, was applied for the purpose of developing, improving and optimizing the processes [36,37,38]. The prediction and optimization capabilities of the RSM are highly appreciated [39]. In this paper, the Box–Behnken design (BBD) of the RSM was selected to arrange the experiments and analyze the experimental data. The details of the input parameters and their ranges were determined by the actual production and reference values in the literature, which are shown in Table 1. The feed speed and cutting length were fixed for all tests and were held constant at 5 m/min and 150 mm, respectively. Each group of experiment was repeated five times and the mean values were used for data analysis and modeling.

Quadratic, linear and 2 FI are frequently employed to model the relationship between input parameters and response parameters. The quadratic model is shown in Equation (1).
(1)$${Y=b}_{0}+\sum_{i=1}^{k}b_{i}X_{i\;}+\sum_{i,j}^{k}b_{\mathrm{ij}}X_{i}X_{j\;}+\sum_{i=1}^{k}b_{\mathrm{ii}}X_{i}^{2}$$
where *Y* represents the response parameters; b_0_ represents a constant coefficient; and b_i_, b_ii_ and b_ij_ represent coefficients of linear, quadratic and interacting terms, respectively.

The response parameters in this study included cutting power and power efficiency, which are defined and shown in Equations (2) and (3). In Figure 2, the schematic diagram of dynamic power change at different cutting stages in one set of experiments is shown.
(2)$$P_{c}{\;=\;P}_{t\;}-{\;P}_{0}$$
(3)$$\;\eta =\frac{P_{c}}{P_{t}}=\frac{P_{c}}{P_{c}{+P}_{0}}=\frac{1}{1+\frac{P_{0}}{P_{c}}}$$
where *P*_c_ is the cutting power (W), *η* is power efficiency, *P*_t_ is the total power during the PB milling process (W) and *P*_0_ is the power during the no-load operation stage of the CNC machine (W).

## 3. Results and Discussion

All the experiments were conducted by the “Run” order shown in Table 2 to avoid the effects of the test sequence on the results’ accuracy. The cutting power and power efficiency were calculated by Equations (2) and (3) and their results are also recorded in Table 2.

### 3.1. Influence of Input Parameters on P_c_ and η

To investigate the variation in power consumption during the PB milling process, the influences of the input parameters on *P*_c_ and *η* were analyzed. Figure 3 shows that *P*_c_ increased with the increase in the rotation speed of the main shaft and depth of milling. However, it decreased with the increase in the rake angle. A decrease in the cutting force has been observed with the increase in the rake angle [40,41,42], resulting in lower cutting power required [43]. With the increase in the depth of milling, the material removal rate increased, which consumed more energy.

PBs, which are made from a mixture of adhesive and shavings, are a heterogeneous material. The impact force between the milling tool and PB material increases as the rotation speed increases, which also causes an increase in the cutting power. From the plot of interaction impact trend (as shown in Figure 4), it is easy to see that a higher cutting power was required when milling with the combination of a higher depth of milling, a higher rotation speed of the main shaft and a lower rake angle.

The power efficiency was found to increase with the increase in the depth of milling and rotation speed of the main shaft. However, the value of *η* decreased with the increase in the rake angle due to reduced cutting force, resulting in reduced energy consumption. The higher values of *η* meant that a higher ratio of energy was consumed in the removal of material. In the Equation (3), *P*_0_ remains constant when the cutting motion parameters are fixed, while the value of *η* increases with the increase in *P*_c_. This explains why the value of *η* increases with the increase in the depth of milling, but decreases with the increase in the rake angle. As shown in Figure 5, the highest value of *η* was achieved by the combination of the highest rotation speed of the main shaft, the highest depth of milling and the lowest rake angle. When the rotation speed of main shaft increased, both *P*_c_ and *P_0_* increased. Hence, the *η* change was not evident.

### 3.2. Analysis of Variance

The analysis of variance (ANOVA) is a widely used technique to investigate the significance of input parameters and evaluate a models’ adequacy. In ANOVA tables, the *p*-value indicates the significance of a factor to a confidence level of 95% and the higher F-value indicates a relatively greater importance of that factor. In Table 3 and Table 4, the *p*-values of the quadratic models were less than 0.0001, indicating that these two models were extremely significant. For the same reason, all of the input parameters exhibited a significant influence on $P_{c}$ and η and many of the interaction terms also had significant influence. For the RSM quadratic models for $P_{c}$ and η, the F-value analysis revealed the depth of milling as the most important factor, followed by the rotation speed of the main shaft and then the rake angle. The highest percentage contribution was exerted by the depth of milling, which also meant the depth of milling was the most important factor [39].

### 3.3. Regression Models

To create the relationship between the input parameters and response parameters, quadratic models were selected over others due to their higher R^2^ values (Table 5). These values reflected that the models were adequate to predict the cutting power and power efficiency, which is demonstrated by the plot of “predicted vs. actual” (Figure 6). The detailed models for *P*_c_ and *η* in terms of coded factors are shown in Equations (4) and (5).
(4)$$P_{c}=39.20-2.77\times A+7.31\times B+12.41\times C+0.275\times A\times B+1.92\times A\times C+0.95\times B\times C+2.07\times A^{2}+5.55\times B^{2}+3.80\times C^{2}$$
(5)$$\eta \;=\;10.33-0.3793\times A+0.4813\times B+2.84\times C+0.3086\times A\times B+0.45\times A\times C+0.3288\times B\times C-0.0771\times A^{2}-0.1983\times B^{2}-0.3069\times C^{2}$$

### 3.4. Optimization of Milling Parameters for PBs

To achieve the purposed energy savings and improved power efficiency during PB milling processes, the lowest $P_{c}$ and highest η would be expected. Even the highest rake angle would be beneficial for decreasing cutting power, as too large of a rake angle would cause the milling tool to rapidly wear, which would have a negative impact on its working life. A higher depth of milling improves the material remove rate, which directly affects milling efficiency. Hence, the detailed conditions of optimization were confirmed and are shown in Table 6. The optimized parameters for $P_{c}$ and η were 2°, 6991.7 rpm, 1.36 mm for rake angle, rotation speed of the main shaft and depth of milling, respectively. The model further predicted that the optimized parameters combination would achieve $P_{c}$ and η values of 52.4 W and 11.9%, respectively, with the desirability of 0.732. To verify the accuracy of the optimization, a validation test was performed with the parameter combination of 2°, 6992 rpm, 1.4 mm for rake angle, rotation speed of the main shaft and depth of milling, respectively. In the validation test, the results of $P_{c}$ and η were 53.2 W and 13.1 %, respectively.

## 4. Conclusions

Power consumption is a key index in material processing which has a direct influence on the economic and environmental benefits of manufacturing processes. The selected input parameters were shown to have significant influences on the response parameters of *P*_c_ and *η*. Quadratic models were developed to model the relationship between input parameters and response parameters. Detailed conclusions are as follows:(1)The value of *P*_c_ increased with the increase in the rotation speed of the main shaft and the depth of milling, but decreased as the rake angle increased. The influence of the input parameters on *η* was similar to that on *P*_c_. The depth of milling was the most important factor for the cutting power and power efficiency during PB milling, followed by the rotation speed of the main shaft and then the rake angle.(2)The values of the regression coefficient, for the *P*_c_ and *η* models, respectively, were 0.9926 and 0.9946 and these values reflected that the quadratic models accurately predicted the values of *P*_c_ and *η*.(3)Higher material removal rates consumed more cutting energy, which had a positive effect on improving power efficiency.(4)In this study, the optimized parameters for *P*_c_ and *η* were 2°, 6992 rpm, 1.4 mm for rake angle, rotation speed of the main shaft and depth of milling, respectively.

## Figures and Tables

**Figure 1 materials-15-00879-f001:**
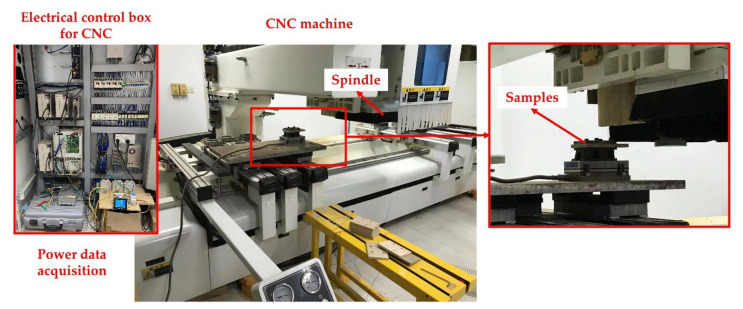
The schematic diagram of experimental setup.

**Figure 2 materials-15-00879-f002:**
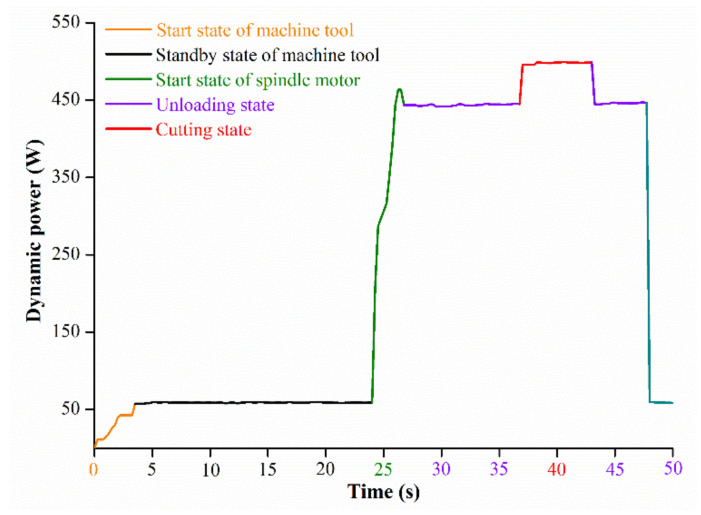
Spindle power plot for a single experiment.

**Figure 3 materials-15-00879-f003:**
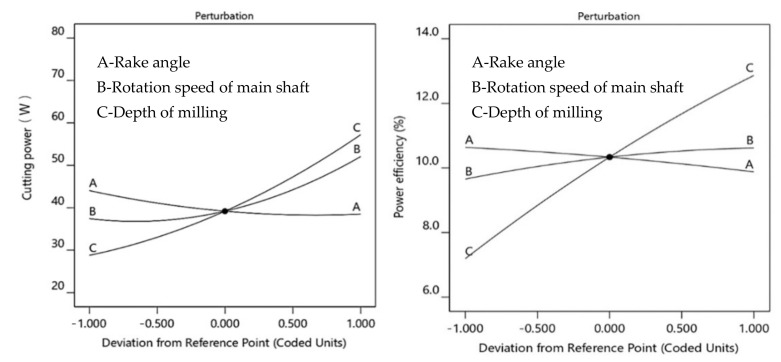
The plot of impact trend for cutting power and power efficiency.

**Figure 4 materials-15-00879-f004:**
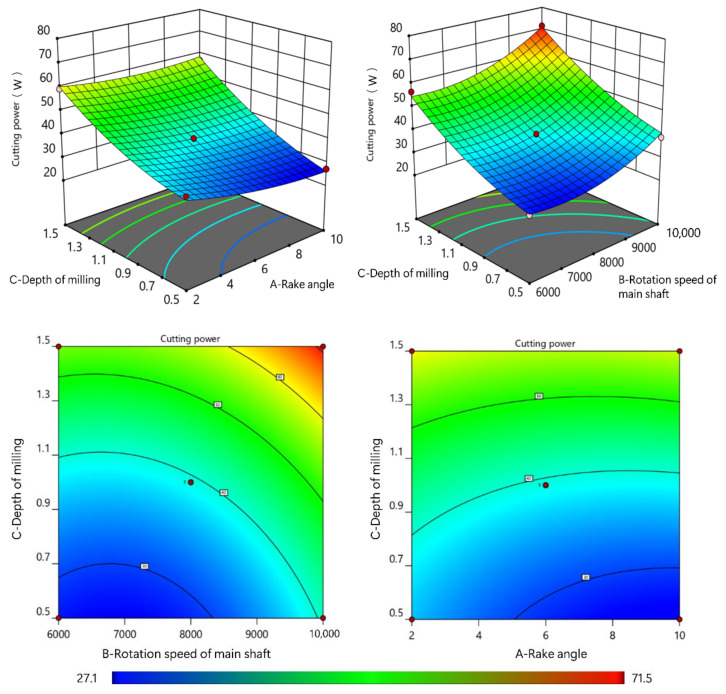
The plot of interaction impact trend for cutting power.

**Figure 5 materials-15-00879-f005:**
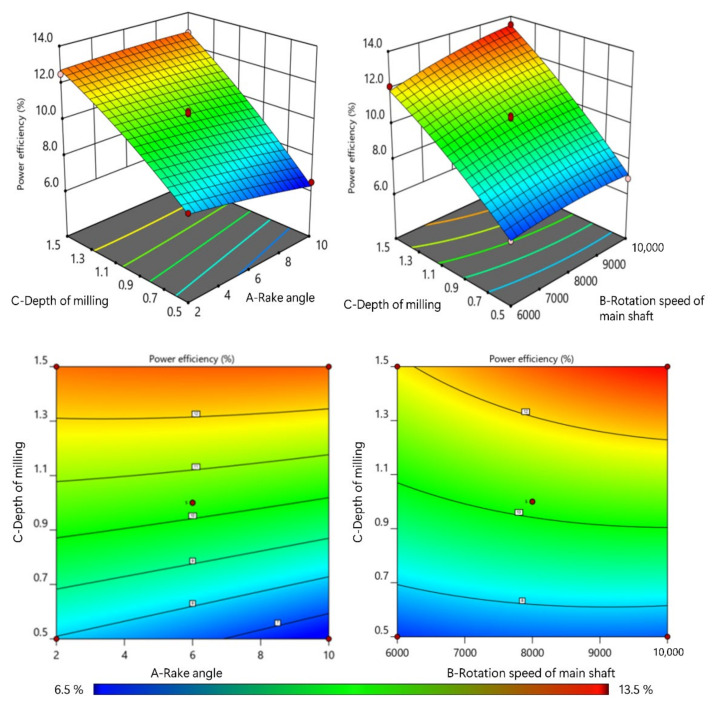
The plot of interaction impact trend for power efficiency.

**Figure 6 materials-15-00879-f006:**
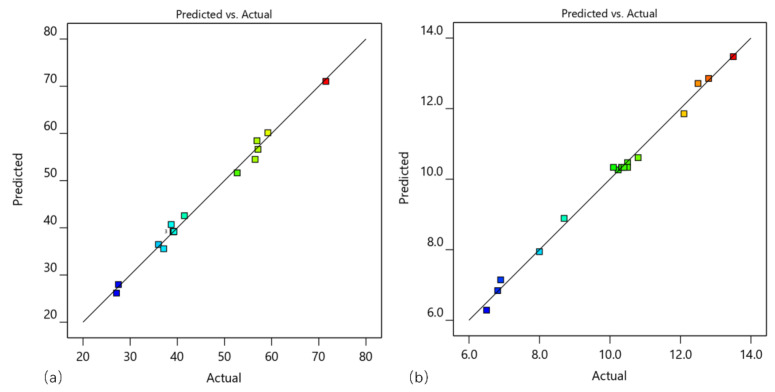
The plot of predicted vs. actual for (**a**) *P*_c_ and (**b**) *η*.

**Table 1 materials-15-00879-t001:** The input parameters and their ranges.

Parameters	Codes	Ranges		
		−1	0	1
Rake angle (°)	A	2	6	10
Rotation speed of main shaft (rpm)	B	6000	8000	10,000
Depth of milling (mm)	C	0.5	1.0	1.5

**Table 2 materials-15-00879-t002:** The results of cutting power and power efficiency during the PB milling process.

Standard	Run	Factors			$P_{c}$ (W)	η (%)
		Rake Angle (°)	Rotation Speed of Main Shaft (rpm)	Depth of Milling (mm)		
1	6	2	6000	1.0	41.5	10.2
2	16	10	6000	1.0	36.0	8.7
3	1	2	10,000	1.0	57.1	10.8
4	14	10	10,000	1.0	52.7	10.5
5	4	2	8000	0.5	37.1	8.0
6	5	10	8000	0.5	27.1	6.5
7	11	2	8000	1.5	59.2	12.5
8	2	10	8000	1.5	56.9	12.8
9	17	6	6000	0.5	27.5	6.8
10	12	6	10,000	0.5	38.7	6.9
11	7	6	6000	1.5	56.5	12.1
12	9	6	10,000	1.5	71.5	13.5
13	10	6	8000	1.0	39.2	10.1
14	8	6	8000	1.0	39.1	10.3
15	3	6	8000	1.0	39.2	10.5
16	13	6	8000	1.0	39.2	10.3
17	15	6	8000	1.0	39.3	10.4

**Table 3 materials-15-00879-t003:** ANOVA results for *P*_c_.

Source	SS	% Cont.	df	MS	F-Value	*p*-Value
Model	2352.99	99.26	9	261.44	103.73	<0.0001
A—Rake angle	61.61	2.60	1	61.61	24.44	0.0017
B—Rotation speed of main shaft	427.78	18.04	1	427.78	169.73	<0.0001
C—Depth of milling	1615.96	68.17	1	1615.96	641.16	<0.0001
AB	0.3025	0.01	1	0.3025	0.1200	0.7392
AC	14.82	0.63	1	14.82	5.88	0.0457
BC	3.61	0.15	1	3.61	1.43	0.2703
A^2^	18.13	0.76	1	18.13	7.19	0.0314
B^2^	129.69	5.47	1	129.69	51.46	0.0002
C^2^	60.80	2.56	1	60.80	24.12	0.0017
Error	37.93	1.60	7			
Cor Total	2370.63	100	16			

SS—Sum of squares MS—Mean square.

**Table 4 materials-15-00879-t004:** ANOVA results for *η*.

Source	SS	% Cont.	df	MS	F-Value	*p*-Value
Model	69.59	99.46	9	7.73	142.02	<0.0001
A—Rake angle	1.15	1.64	1	1.15	21.14	0.0025
B—Rotation speed of main shaft	1.85	2.64	1	1.85	34.04	0.0006
C—Depth of milling	64.32	91.93	1	64.32	1181.49	<0.0001
AB	0.3809	0.54	1	0.3809	7.00	0.0332
AC	0.8100	1.16	1	0.8100	14.88	0.0062
BC	0.4325	0.62	1	0.4325	7.94	0.0258
A^2^	0.0251	0.04	1	0.0251	0.4602	0.5193
B^2^	0.1656	0.24	1	0.1656	3.04	0.1247
C^2^	0.3965	0.57	1	0.3965	7.28	0.0307
Error	0.4394	1.60	7			
Cor Total	69.97	100	16			

SS—Sum of squares MS—Mean square.

**Table 5 materials-15-00879-t005:** Results of ANOVA for different models.

Response Parameters	Models	SD	R^2^	Adj. R^2^	Pred. R^2^	
$P_{c}$	Linear	4.52	0.8881	0.8623	0.8246	
	2FI	4.97	0.8960	0.8336	0.7101	
	Quadratic	1.59	0.9926	0.9830	0.8810	Suggested
η	Linear	0.4508	0.9622	0.9535	0.9273	
	2FI	0.3192	0.9854	0.9767	0.9517	
	Quadratic	0.2333	0.9946	0.9876	0.9307	Suggested

**Table 6 materials-15-00879-t006:** Goals and parameter range for optimization of PB milling process.

Conditions	Goal	Lower Limit	Upper Limit
A	minimize	2	10
B	in range	6000	10,000
C	maximize	0.5	1.5
$P_{c}$	minimize	27.1	71.5
η	maximize	6.5	13.5

## Data Availability

Data are contained within the article.

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
