# Peer review of "Study of Cutting Power and Power Efficiency during Straight-Tooth Cylindrical Milling Process of Particle Boards"

_materials, 2022, doi:10.3390/ma15030879_

Round 1

Reviewer 1 Report

The reviewer comments of the paper

«Study on the Cutting Power and Power Efficiency during Straight Tooth Cylindrical Milling Process of Particle Board»

- Reviewer

The authors presented an article «Study on the Cutting Power and Power Efficiency during Straight Tooth Cylindrical Milling Process of Particle Board». However, there are several points in the article that require further explanation.

Comment 1:

The abstract needs to be improved.

Demonstrate in the abstract novelty, practical significance. Add quantitative and qualitative work results to the abstract.

Comment 2:

The introduction needs to be improved.

Now the list of references needs to be supplemented with at least 3-4 more references published over the past 5 years. Consider the work of the authors: Bustillo A., Chuchala D., Pimenov D.Y., etc.

Add scientific novelty and practical value. Add a clear purpose to the article.

After analyzing the literature, show before formulating the goal of the "blank" spots. Which has not been previously done by other researchers. You must show the importance of the research being undertaken. Show what will be the new research approach in this article. You need to show a hypothesis.

Comment 3:

  1. Materials and Methods

Are all figures original? If not needed appropriate citations and permissions. Refine this for figures throughout the article.

Show the direction of the machine axes. How does this compare to measured cutting forces? What kind of milling scheme is used? Describe in the text.

Describe the measurement procedure in more detail. At what point in time? How is the measuring setup set up? How many repetitions of measurements? What statistical methods are used to process experimental results? Describe the experimental stand in more detail. What method of experiment planning is used and why?

Comment 4:

  1. Results and discussion

On Figures 4 and 5, sign the full axis names. Name figures a, b, c, etc. The reader should not be aware of this.

Comment 5:

It will be useful to add a section of Nomenclature in which to sign all the physical quantities and abbreviations encountered in the article. There are many physical quantities in the text and such a section will help to find the description of the necessary element.

For example,

P                : Cutting power (W)

ANN         : artificial neural network

etc.

Use "rpm" instead of "r/min".

Comment 6:.

It is necessary to more clearly show the novelty of the article and the advantages of the proposed method. Add qualitative and quantitative results of your work. What is the difference from previous work in this area? Show practical relevance.

The article is interesting, but needs to be improved. Authors should carefully study the comments and make improvements to the article step by step. After major changes can an article be considered for publication in the "Materials".

Author Response

- Reviewer 1

The authors presented an article «Study on the Cutting Power and Power Efficiency during Straight Tooth Cylindrical Milling Process of Particle Board». However, there are several points in the article that require further explanation.

Comment 1:

The abstract needs to be improved.

Demonstrate in the abstract novelty, practical significance. Add quantitative and qualitative work results to the abstract.

Dear editor, thanks for your comments. We have revised the text as your suggestions. Hope it could be accepted now.

Comment 2:

The introduction needs to be improved.

Now the list of references needs to be supplemented with at least 3-4 more references published over the past 5 years. Consider the work of the authors: Bustillo A., Chuchala D., Pimenov D.Y., etc.

Add scientific novelty and practical value. Add a clear purpose to the article.

After analyzing the literature, show before formulating the goal of the "blank" spots. Which has not been previously done by other researchers. You must show the importance of the research being undertaken. Show what will be the new research approach in this article. You need to show a hypothesis.

We have added some references from the authors of Bustillo A., Chuchala D., Pimenov D.Y., etc.

Until now, the influences of milling parameters on cutting power and power efficiency and the explicitly quantifying relationships between milling parameters and response parameters were not revealed by any published works. In this paper, the influences of input parameters on response parameters and the explicitly quantifying relationships between milling parameters and response parameters were revealed. It provides a way to create the relationship between milling parameters and cutting power, which is useful for selecting reasonable milling parameters for particle board, to reduce cutting power.

Comment 3:

  1. Materials and Methods

Are all figures original? If not needed appropriate citations and permissions. Refine this for figures throughout the article.

Show the direction of the machine axes. How does this compare to measured cutting forces? What kind of milling scheme is used? Describe in the text.

Describe the measurement procedure in more detail. At what point in time? How is the measuring setup set up? How many repetitions of measurements? What statistical methods are used to process experimental results? Describe the experimental stand in more detail. What method of experiment planning is used and why?

All the figures in the manuscript were original. The direction of machine axes is perpendicular to the horizontal plane.

In this study, peripheral milling was applied. The milling process was same to the experiments for cutting force testing, and the milling direction was along the length direction of workpiece. The particleboard test piece was fixed on the work table of CNC machining center. The dynamic signal of spindle power was acquired by using a three-phase power analyzer with sampling frequency of 50 KHz (AN87300, Ainuo Co., Ltd., Jinan, China). As given in Figure 2, the dynamic cutting power was acquired at the stable cutting state (in red line), namely in the stable cutting process. Each set of experiment was repeated five times.

The response surface methodology was applied to arrange the experiments and analyze the experimental data. The detailed experimental design matrix was listed in Table 2.

Comment 4:

  1. Results and discussion

On Figures 4 and 5, sign the full axis names. Name figures a, b, c, etc. The reader should not be aware of this.

We have revised the Figures 4 and 5.

Comment 5:

It will be useful to add a section of Nomenclature in which to sign all the physical quantities and abbreviations encountered in the article. There are many physical quantities in the text and such a section will help to find the description of the necessary element.

For example,

P                : Cutting power (W)

ANN         : artificial neural network

etc.

Use "rpm" instead of "r/min".

We have added a list of symbols before introduction of manuscript.

Comment 6:

It is necessary to more clearly show the novelty of the article and the advantages of the proposed method. Add qualitative and quantitative results of your work. What is the difference from previous work in this area? Show practical relevance.

We have added the qualitative and quantitative results in abstract and text in the manuscript. Some text for the novelty of the article and the advantages of the proposed method was also added in the revised version.

In the research field of PB processing, the influences of milling parameters on cutting power and power efficiency and the explicitly quantifying relationships between milling parameters and response parameters were not revealed. The RSM, with its advantages of good modeling effect and strong ability to solve multivariable problems, was applied to analyze the experiment data and reveal the effects of input parameters on cutting power and power efficiency. And these created models are useful for selecting milling parameters for particle board, to reduce cutting power.

The article is interesting, but needs to be improved. Authors should carefully study the comments and make improvements to the article step by step. After major changes can an article be considered for publication in the "Materials".

Reviewer 2 Report

The paper presents a study on the Cutting Power and Power Efficiency during the Straight Tooth Cylindrical Milling Process of Particle Board. The topic is relevant and the methodology adopted is appropriate.

  1. The literature part can be improved by comparing the role of machining vibration over Cutting Power and Power Efficiency. You can refer following recent references:
  • A machine learning approach for multipoint tool insert health prediction on VMC, Measurement, Elsevier, vol. 173, no. 108649, pp. 1-16
  • Milling cutter condition monitoring using machine learning approach, Materials Science and Engineering, IOP Publishing Ltd, UK, vol. 624, no. 1, pp. 1-7
  • Review on tool condition classification in milling: A machine learning approach, Materials Today: Proceedings, vol. 46, no. 2, pp. 1106-1115
  • Deep Learning Algorithms for Tool Condition Monitoring in Milling: A Review, Journal of Physics: Conference Series, IOP Publishing, vol. 1969, no. 012039
  1. What is the meaning of short-form ‘PB’? You must define ‘PB’ when used in the first place.
  2. The sentence “PB materials were supplied by Dare Global Technology Group (Danyan Jiangsu, 84 China)” is irrelevant and must be removed. You can add this part in acknowledgment.
  3. Mentioning “Nanxing Ma- 91 chinery Co. Ltd., Dongguan, China” is irrelevant and must be removed. You can add this part in acknowledgment.
  4. Mentioning “AN87300, Ainuo Co., Ltd, Shandong, China” is irrelevant and must be removed. You can add this part in acknowledgment.
  5. Figure 1 must be an actual real-time experimental setup where the CNC machine, the exact location of sensors, and its integration of data acquisition system must be shown. The current picture is so discrete.
  6. Authors must clearly highlight contributions and innovation of methods & principles.
  7. Why is ANOVA preferred over the machine learning model?
  8. The spelling of “angle” is written as “angel”. Check spelling mistakes throughout the manuscript.
  9. What were the criteria behind selecting various input parameters such as “rake angle, speed, depth, etc.?”
  10. What must be the reason behind the superior fitting of the “Quadratic” model over the other two? You must justify it with respect to the effect of variation in input parameters on Cutting Power and Power Efficiency.
  11. It is suggested to perform the parameter importance analysis that affects Cutting Power and Power Efficiency.

I recommended that this submission be reconsidered for review contingent revisions suggested.

Author Response

Reviewer 2

The paper presents a study on the Cutting Power and Power Efficiency during the Straight Tooth Cylindrical Milling Process of Particle Board. The topic is relevant and the methodology adopted is appropriate.

  1. The literature part can be improved by comparing the role of machining vibration over Cutting Power and Power Efficiency. You can refer following recent references:
  • A machine learning approach for multipoint tool insert health prediction on VMC, Measurement, Elsevier, vol. 173, no. 108649, pp. 1-16
  • Milling cutter condition monitoring using machine learning approach, Materials Science and Engineering, IOP Publishing Ltd, UK, vol. 624, no. 1, pp. 1-7
  • Review on tool condition classification in milling: A machine learning approach, Materials Today: Proceedings, vol. 46, no. 2, pp. 1106-1115
  • Deep Learning Algorithms for Tool Condition Monitoring in Milling: A Review, Journal of Physics: Conference Series, IOP Publishing, vol. 1969, no. 012039

We have added these references in the introduction.

  1. What is the meaning of short-form ‘PB’? You must define ‘PB’ when used in the first place.

We have defined the short-form ‘PB’ in the text and a list of symbols was added in the text.

  1. The sentence “PB materials were supplied by Dare Global Technology Group (Danyan Jiangsu, 84 China)” is irrelevant and must be removed. You can add this part in acknowledgment.

We have removed this information to acknowledgment.

  1. Mentioning “Nanxing Ma- 91 chinery Co. Ltd., Dongguan, China” is irrelevant and must be removed. You can add this part in acknowledgment.

We have removed this information to acknowledgment.

  1. Mentioning “AN87300, Ainuo Co., Ltd, Shandong, China” is irrelevant and must be removed. You can add this part in acknowledgment.

We have removed this information to acknowledgment.

  1. Figure 1 must be an actual real-time experimental setup where the CNC machine, the exact location of sensors, and its integration of data acquisition system must be shown. The current picture is so discrete.

We have revised the Figure 1.

  1. Authors must clearly highlight contributions and innovation of methods & principles.

We have revised the text in abstract and added quantitative and qualitative work results to the abstract. And we described the contributions and innovation of work in the revised manuscript.

  1. Why is ANOVA preferred over the machine learning model?

It is a nice suggestion. Machine learning model is a kind of new model with some advantages. However, we used experimental method of response surface methodology in this study. ANOVA was applied to analyze the model significance. Machine learning model will be used as a new technical in our future research.

  1. The spelling of “angle” is written as “angel”. Check spelling mistakes throughout the manuscript.

We have revised these spelling in the text.

  1. What were the criteria behind selecting various input parameters such as “rake angle, speed, depth, etc.?”

The choice of parameter levels is based on actual production and available literature.

  1. What must be the reason behind the superior fitting of the “Quadratic” model over the other two? You must justify it with respect to the effect of variation in input parameters on Cutting Power and Power Efficiency.

The detailed results of ANOVA were shown as following figures. The values of quadratic models were higher than the other kind of models. The larger value of R2 is, the better the fitting effect of the model is. That is the main reason why the “Quadratic” models were chosen. The following references also prove the correctness of this choice. In the validation test, the results were very close to the model prediction values, which also meant the “Quadratic” model is competent to predict the cutting power and power efficiency during PB milling process.

Wang, H.; Huang, L.; Cao, P.; Ji, F.; Yang, G.; Guo, X.; Li, R. Investigation of Shear Strength of Engineered Wood Flooring Bonded with PUR by Response Surface Methodology. Bioresources 2017, 12, 3656-3665, doi:10.15376/biores.12.2.3656-3665.

Zhu, Z.; Buck, D.; Cao, P.; Guo, X.; Wang, J. Assessment of Cutting Forces and Temperature in Tapered Milling of Stone–Plastic Composite Using Response Surface Methodology. JOM 2020, 72, 3917-3925.

Valarmathi, T.N.; Palanikumar, K.; Sekar, S. Modeling of thrust force in drilling of plain medium density fiberboard (MDF) composite panels using RSM. In International Conference on Modelling Optimization and Computing, Rajesh, R., Ganesh, K., Koh, S.C.L., Eds. 2012; Vol. 38, pp. 1828-1835.

Bhushan, R.K. Optimization of cutting parameters for minimizing power consumption and maximizing tool life during machining of Al alloy SiC particle composites. J Clean Prod 2013, 39, 242-254, doi:10.1016/j.jclepro.2012.08.008.

  1. It is suggested to perform the parameter importance analysis that affects Cutting Power and Power Efficiency.

We have added some analysis in the manuscript.

For RSM quadratic models for Pc and η, the F-value analysis reveals the depth of milling as the most important factor followed by the rotation speed of main shaft and then the rake angle. The highest percentage contribution is exerted by depth of milling, which also meant the depth of milling is the most important factor.

I recommended that this submission be reconsidered for review contingent revisions suggested.

Round 2

Reviewer 1 Report

The authors have improved the article in accordance with the comments. Now the article can be accepted for publication.

Reviewer 2 Report

The authors have addressed most of my comments and have improved the paper's quality. Therefore, in my opinion, the current manuscript fits the standards to be published, and the presentation of the problem at hand is clear. In addition, this study has interest from an academic viewpoint. Therefore, the current manuscript can be accepted for publication.